# Prevalence and correlates of restless legs syndrome in men living with HIV

**Douglas M. Wallace**[1,2]*, **Maria L. Alcaide**[3], **William K. Wohlgemuth**[4], **Deborah L. Jones Weiss**[5], **Claudia Uribe Starita**[3], **Sanjay R. Patel**[6], **Valentina Stosor**[7], **Andrew Levine**[8], **Carling Skvarca**[9], **Dustin M. Long**[10], **Anna Rubtsova**[11], **Adaora A. Adimora**[12], **Stephen J. Gange**[13], **Amanda B. Spence**[14], **Kathryn Anastos**[15], **Bradley E. Aouizerat**[16], **Yaacov Anziska**[17], **Naresh M. Punjabi**[3]

1 Neurology Service, Bruce W. Carter Department of Veterans Affairs Medical Center, Miami, Florida, United States of America, 2 Department of Neurology, University of Miami Miller School of Medicine, Miami, Florida, United States of America, 3 Department of Medicine, University of Miami Miller School of Medicine, Miami, Florida, United States of America, 4 Psychology Service, Bruce W. Carter Department of Veterans Affairs Medical Center, Miami, Florida, United States of America, 5 Department of Psychiatry and Behavioral Sciences, University of Miami Miller School of Medicine, Miami, Florida, United States of America, 6 Division of Pulmonary, Allergy and Critical Care Medicine, University of Pittsburgh, Pittsburgh, Pennsylvania, United States of America, 7 Department of Medicine and Surgery, Northwestern University Feinberg School of Medicine, Chicago, Illinois, United States of America, 8 Department of Neurology, David Geffen School of Medicine, University of California Los Angeles, Los Angeles, California, United States of America, 9 Department of Medicine, University of Pittsburgh, Pittsburgh, Pennsylvania, United States of America, 10 Department of Biostatistics, School of Public Health, University of Alabama at Birmingham, Birmingham, Alabama, United States of America, 11 Department of Behavioral, Social, and Health Education Sciences, Emory University Rollins School of Public Health, Atlanta, Georgia, United States of America, 12 Department of Medicine, University of North Carolina School of Medicine, University of North Carolina at Chapel Hill, Chapel Hill, North Carolina, United States of America, 13 John Hopkins Bloomberg School of Public Health, Baltimore, Maryland, United States of America, 14 Division of Infectious Disease, Georgetown University Medical Center, Washington, District of Columbia, United States of America, 15 Department of Medicine and Epidemiology and Population Health, Albert Einstein College of Medicine and Montefiore Health System, Bronx, New York, United States of America, 16 Bluestone Center for Clinical Research, College of Dentistry, New York University, New York, New York, United States of America, 17 Department of Neurology, State University of New York-Downstate Medical Center, Brooklyn, New York, United States of America

* dwallace@med.miami.edu

**Data Availability Statement:** All data are the property of MACS/WIHS Combined Cohort Study Data Access (MWCCS) and cannot be publicly shared. Access to individual-level data from

## Abstract

### Background

Data on the prevalence and correlates of restless legs syndrome (RLS) in people with HIV are limited. This study sought to determine the prevalence of RLS, associated clinical correlates, and characterize sleep-related differences in men with and without HIV.

### Methods

Sleep-related data were collected in men who have sex with men participating in the Multicenter AIDS Cohort Study (MACS). Demographic, health behaviors, HIV status, comorbidities, and serological data were obtained from the MACS visit coinciding with sleep assessments. Participants completed questionnaires, home polysomnography, and wrist actigraphy. RLS status was determined with the Cambridge-Hopkins RLS questionnaire. RLS prevalence was compared in men with and without HIV. Multinomial logistic regression

MWCCS may be obtained upon review and approval of a MWCCS concept sheet. Links and instructions for online concept sheet submission are on the study website (https://statepi.jhsph.edu/mwccs/work-with-us/).

**Funding:** The contents of this publication are solely the responsibility of the authors and do not represent the official views of the National Institutes of Health (NIH). MWCCS (Principal Investigators): Atlanta CRS (Ighovwerha Ofotokun, Anandi Sheth, and Gina Wingood), U01-HL146241; Baltimore CRS (Todd Brown and Joseph Margolick), U01-HL146201; Bronx CRS (Kathryn Anastos and Anjali Sharma), U01-HL146204; Brooklyn CRS (Deborah Gustafson and Tracey Wilson), U01-HL146202; Data Analysis and Coordination Center (Gypsyamber D'Souza, Stephen Gange and Elizabeth Golub), U01-HL146193; Chicago-Cook County CRS (Mardge Cohen and Audrey French), U01-HL146245; Chicago-Northwestern CRS (Steven Wolinsky), U01-HL146240; Northern California CRS (Bradley Aouizerat, Jennifer Price, and Phyllis Tien), U01-HL146242; Los Angeles CRS (Roger Detels and Matthew Mimiaga), U01-HL146333; Metropolitan Washington CRS (Seble Kassaye and Daniel Merenstein), U01-HL146205; Miami CRS (Maria Alcaide, Margaret Fischl, and Deborah Jones), U01-HL146203; Pittsburgh CRS (Jeremy Martinson and Charles Rinaldo), U01-HL146208; UAB-MS CRS (Mirjam-Colette Kempf, Jodie Dionne-Odom, and Deborah Konkle-Parker), U01-HL146192; UNC CRS (Adaora Adimora), U01-HL146194. The MWCCS is funded primarily by the National Heart, Lung, and Blood Institute (NHLBI), with additional co-funding from the Eunice Kennedy Shriver National Institute Of Child Health & Human Development (NICHD), National Institute On Aging (NIA), National Institute Of Dental & Craniofacial Research (NIDCR), National Institute Of Allergy And Infectious Diseases (NIAID), National Institute Of Neurological Disorders And Stroke (NINDS), National Institute Of Mental Health (NIMH), National Institute On Drug Abuse (NIDA), National Institute Of Nursing Research (NINR), National Cancer Institute (NCI), National Institute on Alcohol Abuse and Alcoholism (NIAAA), National Institute on Deafness and Other Communication Disorders (NIDCD), National Institute of Diabetes and Digestive and Kidney Diseases (NIDDK), National Institute on Minority Health and Health Disparities (NIMHD), and in coordination and alignment with the research priorities of the National Institutes of Health, Office of AIDS Research (OAR). MWCCS data collection is also supported by UL1-TR000004 (UCSF CTSA), UL1-TR003098 (JHU ICTR), UL1-TR001881

was used to examine correlates of RLS among all participants and men with HIV alone. Sleep-related differences were examined in men with and without HIV by RLS status.

## Results

The sample consisted of 942 men (56% HIV+; mean age 57 years; 69% white). The prevalence of definite RLS was comparable in men with and without HIV (9.1% vs 8.7%). In multinomial regression, HIV status was not associated with RLS prevalence. However, white race, anemia, depression, and antidepressant use were each independently associated with RLS. HIV disease duration was also associated with RLS. Men with HIV and RLS reported poorer sleep quality, greater sleepiness, and had worse objective sleep efficiency/fragmentation than men without HIV/RLS.

## Conclusions

The prevalence of RLS in men with and without HIV was similar. Screening for RLS may be considered among people with HIV with insomnia and with long-standing disease.

## Introduction

Restless legs syndrome (RLS) or Willis-Ekbom disease is a neurological disorder which interferes with rest and sleep, leading to impaired daytime functioning and decreased quality of life [1]. The cardinal symptoms consist of an irresistible urge to move the limbs, occurring after periods of inactivity, worsening at night, that improve with moving the extremities [2]. Symptoms of RLS can prevent people from falling asleep [3]. In addition, up to 90% of people with RLS may also experience periodic limb movements of sleep (PLMS), which are repetitive flexion movements of the toe, ankle, and hip that can further fragment sleep [1, 4]. Among the US general population, the prevalence of RLS ranges between 2.5–16.0% with a higher prevalence in women, older adults, and those of northern European descent [5–7]. In addition, chronic conditions such as iron deficiency anemia, type 2 diabetes, chronic kidney disease, and major depression have been associated with RLS [8, 9].

Data are sparse about the prevalence or predictors of RLS in people living with human immunodeficiency virus (HIV). To date, only three studies exist examining the prevalence of RLS in people living with HIV (PLHIV) and have been equivocal regarding the independent influence of HIV infection on RLS prevalence [10–12]. Use of clinic-based samples, varying definitions of RLS, limited sample sizes, and lack of assessment of other medical comorbidities (e.g., chronic kidney disease, diabetes) and lifestyle factors (e.g., illicit drug use) are likely contributors to the incongruent findings across available studies [10–12]. Furthermore, there are no data on objective sleep correlates of RLS symptoms in PLHIV derived either from actigraphy or polysomnography. Given that comorbid sleep disorders, such as sleep-disordered breathing and habitually short sleep duration can worsen RLS symptoms, characterizing and accounting for such disorders would allow for an assessment as to whether HIV infection is an independent contributor [13, 14]. Thus, a more comprehensive understanding of the prevalence and determinants of RLS in PLHIV may help determine the burden of disease and potentially customize future intervention studies.

To address such limitations, recently collected data on sleep in the Multicenter AIDS Cohort Study (MACS) were leveraged along with the comprehensive battery of measures

(UCLA CTSI), P30-AI-050409 (Atlanta CFAR), P30-AI-073961 (Miami CFAR), P30-AI-050410 (UNC CFAR), P30-AI-027767 (UAB CFAR), and P30-MH-116867 (Miami CHARM). The funders had no role in study design, data collection and analysis, decision to publish, or preparation of the manuscript.

**Competing interests:** The authors have declared that no competing interests exist.

related to HIV, lifestyle, and medication use to define the prevalence and correlates of RLS in a large, ethnically and racially diverse community-based sample. The aims of this study were to determine the prevalence of RLS in men with and without HIV, examine if viral load, CD4 count, and HIV disease duration are associated with the prevalence and severity of RLS, and assess the level of subjective and objective sleep-related impairment in men with HIV and RLS compared to men without RLS.

## Methods

### Participants

The study sample consisted of 942 men with and without HIV participating in an ancillary sleep protocol of the Multicenter Aids Cohort Study (MACS). Briefly, the MACS is a longitudinal cohort study (1984–2019) of HIV, its treatment trajectories, and associated comorbidities in men who have sex with men [15]. The cohort includes those with HIV and those at risk for HIV. Study participants complete various psychosocial/behavioral questionnaires, and laboratory and physical exam assessments every six months. The demographic, anthropometric, health behaviors, HIV status/disease severity, medical comorbidities and medication variables used for this analysis come from the parent study visits coinciding with the sleep assessments. Participants for the ancillary sleep protocol were recruited from four US sites (Baltimore, Chicago, Pittsburgh, Los Angeles) from April 2018 to September 2019. Study participants who agreed to participate were asked to complete sleep questionnaires and objective sleep testing as described below. To be included in this analysis, participants had to complete the sleep questionnaires, overnight polysomnography, and actigraphy. Participants were excluded if the RLS questionnaire was not completed or the sleep recordings were of poor quality that would have precluded sleep staging or scoring of disordered breathing events. The Investigational Review Board at Johns Hopkins University approved the protocol (IRB00219740) and all participants signed a written informed consent prior to enrollment.

### Subjective sleep measures

Symptoms of RLS were determined per the Cambridge-Hopkins restless legs syndrome questionnaire (CH-RLSq) [16]. This self-reported instrument queries not only for the diagnostic criteria of RLS but also excludes its clinical mimickers. As per instructions of this instrument's developer, three RLS categories were generated by this questionnaire: definite RLS, no RLS, or indeterminate. These CH-RLSq generated categories are used throughout the manuscript. Items embedded in the questionnaires assess age at symptom onset, symptom frequency, and distress level associated with disease. The CH-RLSq has been found to have a sensitivity and specificity of 87% and 94%, respectively, for determining RLS from not RLS [16].

Global sleep quality was measured per the Pittsburgh sleep quality index (PSQI). Individual items assess sleep onset latency, habitual sleep times, time in bed, and sleep duration. A global PSQI score greater than five yields a diagnostic sensitivity of 89.6% and specificity of 86.5% in distinguishing good and poor sleepers [17].

Daytime sleepiness was determined using the Epworth sleepiness scale (ESS) which consists of eight items rated on a scale of 0–3 with higher scores indicating a greater propensity to fall asleep in different situations [18]. An ESS score > 10 was considered a cut off value for excessive daytime sleepiness.

### Objective sleep assessments

A single night of home polysomnography was performed at the participant's home with a type II portable sleep monitor (Nox A1 PSG; Nox Medical, Reykjavik, Iceland). Sleep parameters

were measured using the recommended criteria [19]. Apnea was defined as cessation of airflow for at least 10 seconds. Hypopneas were defined as a 30% reduction in airflow for at least 10 seconds associated with either a 3% oxygen desaturation or arousal. Obstructive sleep apnea (OSA) was diagnosed if the apnea-hypopnea index (AHI) was at least 5 events/hr of sleep. Periodic limb movements were scored per standard criteria [3].

Rest-sleep patterns were assessed over seven consecutive days using actigraphy (Actiwatch Spectrum, Philips Respironics, Murrysville, PA) worn on the dominant wrist. An actigraph is a wristwatch-like device that records movement through an accelerometer and has a photodiode for detecting light intensity exposure. Activity data were collected in 30-second epochs and sleep-wake inferences were made using the Actiware automated scoring algorithm. The wake threshold value (i.e., the number of activity counts used to define wake) was set to medium sensitivity of 40.0 activity counts per epoch. Estimated sleep onset latency, total sleep time, sleep efficiency, and fragmentation index were determined.

## HIV status, health behaviors, and medical comorbidities

The following markers were used as surrogates of HIV disease or treatment. HIV status was determined at study entry by a commercially available HIV test as per study protocol [15]. For men with HIV, viral load (HIV RNA copies/ml), CD4 Tcell count/ml, CD4 Tcell count nadir, and HIV disease duration were recorded. Undetectable viral load was defined as lab value of less than 20 viral copies/ml. Antiretroviral (ART) treatment and cumulative exposure history were obtained from interviewer administered assessments and historical data collected as part of the biannual visits.

Several health behaviors, comorbid conditions, and medications that have been previously reported to be associated with RLS symptoms were examined at visits coinciding with the sleep ancillary study [9]. Smoking status and heavy alcohol consumption (> 14 drinks/week) were based on self-report [20]. Similarly, daily use of marijuana/hash and illicit heroin/opiate use since last visit was also assessed. Hypertension was determined per self-report or use of antihypertensive medication. Type 2 diabetes was determined by fasting glucose $\geq$ 126 mg/dl, hemoglobin $A_1c \geq 6.5\%$, or use of hypoglycemic medications. Chronic kidney disease was defined based on estimated glomerular filtration rate < 60 ml/min [21]. Anemia was diagnosed if the hemoglobin or erythropoietin value was below 5[th] percentile for age [22]. Current depressive symptoms were per Center for Epidemiologic Studies Depression (CESD) scale $\geq$ 16 [23]. Finally, use of medications known to exacerbate (i.e., antidepressants) or ameliorate (i.e., sedatives/anxiolytic/tranquilizers) RLS symptoms was obtained from the biannual visit.

## Data analysis

Demographic and clinical characteristics were compared between men with and without HIV. Student's t-test was used for continuous variables and chi-square test was used for categorical data. The unadjusted prevalence of RLS and characteristics were subsequently compared by HIV status. To examine the independent association between HIV status and prevalent RLS or indeterminate status, multivariable logistic regression models for RLS status were used to adjust for potential confounders. Based on prior research [5, 7, 9], multinomial logistic regression models were constructed containing independent variables in the following domains: sociodemographic (age, race), health behaviors (smoking status, excessive alcohol use, substance use status), and medical comorbidities (HIV, diabetes, hypertension, chronic kidney disease, depressive symptoms). To determine whether HIV-related variables were associated with RLS status, an additional multinomial model was constructed sequentially adjusting for

age and other independent variables found to be significantly associated with RLS status in the initial overall model. Histograms and frequency distributions of the dependent variables (e.g., definite RLS, no RLS, indeterminate) were constructed. Bivariate associations were examined using scatter plots, tabular methods, and analysis of variance as appropriate for the types of variables. Confounding was examined between our primary predictor (e.g., HIV status) and our covariates using correlations for continuous data and cross-tabulations for categorical data. To characterize sleep-related burden by HIV status, responses on sleep questionnaires and objective measures were compared between men with and without RLS. Measures of sleep-disordered breathing and periodic limb movements during sleep were compared among the groups. Similarly, actigraphy-measures of sleep timing, fragmentation, and nocturnal movements were also compared. Comparison statistics was per Student's t-test for continuous data while Chi-square test was used for categorical data. Post-hoc tests were performed with Tukey's test where significant differences between the four groups were detected. For all analyses, $p < 0.05$ was defined as statistically significant. Statistical analyses were performed with SPSS Statistics 25.0 (SPSS, Chicago, IL).

## Results

### Sample characteristics

The sample consisted of 942 middle-aged men (526 HIV positive, 416 HIV negative; 69% white) recruited from four clinical sites (Table 1). Twenty percent of the sample were current smokers, 7% met criteria for excessive alcohol use, and 11% reported daily use of marijuana or hash. Comorbidities were prevalent with 22% having hypertension and 25% reporting depressive symptoms. Laboratory-based criteria diagnoses of type 2 diabetes, chronic kidney disease, and anemia were met for 16%, 16%, and 10% of the cohort, respectively. Among men with HIV, the mean length of HIV infection was 27 ± 8 years with most using effective ART treatment with undetectable HIV viral load (75%) and CD4 counts above 500 (74%). Compared to men without HIV, men with HIV were significantly younger (54 ± 11 vs 60 ± 12 years, $p < 0.001$), more likely to be of racial minority background (37% vs 23%, $p < 0.001$), and more likely to be of Hispanic/Latino ethnicity (18% vs 7%, $p < 0.001$). Furthermore, men with HIV were significantly more likely to be current smokers (24% vs 14%, $p < 0.001$) and meet diagnostic criteria for anemia (13% vs 6%, $p < 0.001$) and chronic kidney disease (20% vs 12%, $p < 0.001$) than men without HIV. No other demographic, comorbidity, or medication use differences were noted between the groups.

### Prevalence and correlates of RLS by HIV status

Fig 1 compares RLS prevalence by HIV status. The prevalence of "definite" RLS was similar among men with HIV (n = 48; 9.1%) and men without HIV (n = 36; 8.7%). However, the prevalence for the "indeterminate" category was higher among men with HIV relative to men without HIV (17.9% vs 12.0%, $p = 0.039$). Comparing RLS characteristics among those with "definite RLS", men with HIV had similar age at onset of symptoms (48 ± 14 vs 47 ± 16 years; $p = 0.79$), levels of moderate to severe distress (63% vs 56%, $p = 0.65$), and frequency of RLS symptoms than men without HIV. Specifically, among men with HIV, 40% reported symptoms once weekly or less often, 31% reported symptoms 2–3 days weekly, and 29% reported symptoms 4 days or more weekly. A similar distribution of symptoms was observed among men without HIV with 31% reporting symptoms once weekly or less frequently, 36% reporting symptoms 2–3 times weekly, and 33% reporting symptoms at least 4 times weekly.

To determine if HIV was associated with "definite" or "indeterminate" RLS status after adjusting for potential confounders, a multinomial logistic regression model was constructed.

**Table 1. Comparison of men with and without HIV by characteristics.**

| Characteristic | All | HIV + | HIV - | p-value |
|---|---|---|---|---|
| | (n = 942) | (n = 526) | (n = 416) | |
| *Demographics* | | | | |
| Age (yrs) | 57 ± 12 | 54 ± 11 | 60 ± 12 | <0.001 |
| BMI (kg/m$^2$) | 27 ± 5 | 27 ± 5 | 28 ± 5 | 0.11 |
| Race, n (%) | | | | <0.001 |
| White | 652 (69) | 332 (63) | 320 (77) | |
| Black | 270 (29) | 182 (35) | 88 (21) | |
| Other | 20 (2) | 12 (2) | 8 (2) | |
| Hispanic ethnicity, n (%) | 125 (13) | 95 (18) | 30 (7) | <0.001 |
| Recruitment site, n (%) | | | | 0.29 |
| Baltimore | 264 (28) | 144 (27) | 120 (29) | |
| Chicago | 199 (21) | 123 (23) | 76 (18) | |
| Pittsburgh | 230 (24) | 123 (23) | 107 (26) | |
| Los Angeles | 249 (26) | 136 (26) | 113 (27) | |
| *Health behaviors, n (%)* | | | | |
| Current smoker | 187 (20) | 127 (24) | 60 (14) | <0.001 |
| Excessive alcohol use | 70 (7) | 33 (6) | 37 (9) | 0.14 |
| Daily marijuana | 100 (11) | 59 (11) | 41 (10) | 0.52 |
| Heroin/Opiates | 32 (3) | 22 (4) | 10 (2) | 0.13 |
| *Comorbidities, n (%)* | | | | |
| Anemia | 95 (10) | 71 (13) | 24 (6) | <0.001 |
| Hypertension | 204 (22) | 98 (18) | 106 (25) | 0.01 |
| Type 2 Diabetes | 151 (16) | 92 (17) | 59 (14) | 0.17 |
| Chronic kidney disease | 155 (16) | 106 (20) | 49 (12) | 0.001 |
| Depression | 235 (25) | 146 (28) | 89 (21) | 0.03 |
| *Medications, n (%)\** | | | | |
| Anxiolytic/sedatives | 180 (22) | 98 (22) | 82 (23) | 0.87 |
| Antidepressants | 147 (18) | 88 (20) | 59 (16) | 0.23 |
| *HIV variables* | | | | |
| Disease duration (yrs) | - | 27 ± 8 | - | |
| CD4 nadir (pre-treatment) | - | 463 ± 244 | - | |
| CD4 count > 500, n (%) | - | 387 (74) | - | |
| Undetectable HIV viral load | - | 394 (75) | - | |
| ART treatment | - | 509 (97) | - | |

Data presented as means ± SD or frequency (%). Group comparisons per Students t-test or Chi-square tests.

\* The medication variable contained missing data with n = 802. BMI body mass index; ART, antiretroviral therapy.

As shown in Table 2, HIV status was not associated with "indeterminate" or "definite" RLS in these models. The only factor significantly associated with "indeterminate" RLS status was current depressive symptoms (OR 1.68; 95% CI 1.07–2.65). In contrast, covariates significantly associated with "definite" RLS included white race (OR 2.20; 95% CI 1.10–4.38), anemia (OR 2.22; 95% CI 1.03–4.79), current depressive symptoms (OR 1.82; 95% CI 1.05–3.15), and active use of antidepressant medications (OR 2.03; 95% CI 1.13–3.64).

To examine whether HIV specific factors were associated with RLS status, a multinomial model was constructed in only those with HIV, sequentially adjusting for age and the significant covariates listed above. As shown in Table 3, none of the three characteristics examined

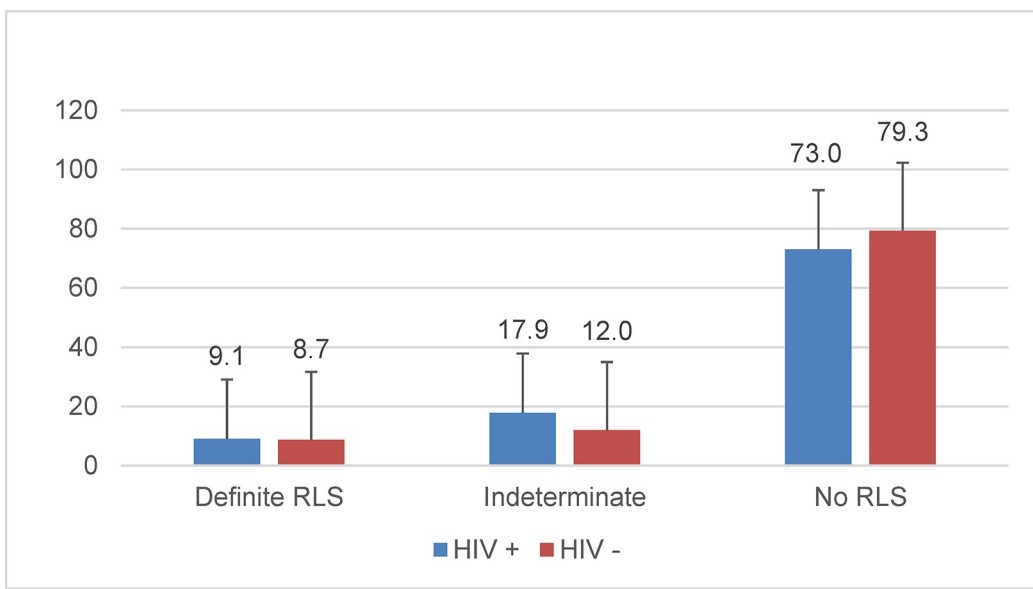

**Fig 1. Unadjusted prevalence of RLS by HIV status.** Data represent RLS status prevalence with error bars.

(HIV duration, CD4 count > 500, or viral load suppression) was associated with "indeterminate" RLS. In contrast, HIV disease duration was associated with "definite" RLS (OR 1.08; 95% CI 1.01–1.16) in the fully adjusted model. To assess the potential effects of disease duration, HIV disease duration and RLS prevalence were cross-tabulated. No cases of "definite RLS"

**Table 2. Adjusted odds ratio§ for RLS status in men with and without HIV.**

| Variables | Indeterminate | | Definite | |
|---|---|---|---|---|
| | RLS (N = 117) | | RLS (N = 74) | |
| *Demographics* | | | | |
| Age | 0.99 | 0.97–1.01 | 1.00 | 0.98–1.03 |
| White race (ref non-white) | 1.45 | 0.88–2.41 | 2.20 | 1.10–4.38 * |
| *Health behaviors* | | | | |
| Current smoker | 1.21 | 0.71–2.07 | 0.99 | 0.49–2.00 |
| Alcohol abuse | 1.47 | 0.73–2.94 | 1.16 | 0.46–2.90 |
| Daily marijuana | 1.23 | 0.65–2.32 | 1.38 | 0.62–3.05 |
| Heroin/Opiates | 0.72 | 0.20–2.56 | 2.75 | 0.97–7.79 |
| *Comorbidities* | | | | |
| HIV | 1.52 | 0.98–2.35 | 1.25 | 0.73–2.13 |
| Anemia | 1.10 | 0.53–2.27 | 2.22 | 1.03–4.79 * |
| Hypertension | 1.36 | 0.84–2.21 | 1.71 | 0.98–3.00 |
| Diabetes | 1.11 | 0.65–1.90 | 1.11 | 0.58–2.10 |
| Chronic kidney disease | 0.93 | 0.54–1.61 | 0.80 | 0.41–1.57 |
| Depression | 1.68 | 1.07–2.65 * | 1.82 | 1.05–3.15 * |
| *Medications* | | | | |
| Anxiolytic/sedatives | 1.02 | 0.61–1.69 | 0.92 | 0.50–1.71 |
| Antidepressants | 1.07 | 0.62–1.84 | 2.03 | 1.13–3.64 * |

§Based on multivariable multinomial logistic regression results (referenced to "no RLS"; n = 587).

* p < 0.05, ** p < 0.01.

**Table 3. Multinomial models of HIV specific predictors of RLS status in men with HIV.**

| HIV variable | Model A | | Model B | | Model C | | Model D | |
|---|---|---|---|---|---|---|---|---|
| | OR | 95% CI | OR | 95% CI | OR | 95% CI | OR | 95% CI |
| Definite RLS (N = 47) | | | | | | | | |
| Disease duration (years) | 1.05 | 1.01–1.10 * | 1.05 | 0.99–1.12 | 1.06 | 1.00–1.13 * | 1.08 | 1.02–1.16 * |
| CD4 count > 500 | 1.64 | 0.73–3.67 | 1.61 | 0.71–3.63 | 1.94 | 0.83–4.54 | 1.53 | 0.65–3.60 |
| Undetectable viral load | 1.70 | 0.33–3.95 | 1.61 | 0.69–3.78 | 1.59 | 0.67–3.77 | 1.75 | 0.69–4.45 |
| Indeterminate RLS (N = 90) | | | | | | | | |
| Disease duration (yrs) | 1.00 | 0.97–1.04 | 1.03 | 0.99–1.07 | 1.03 | 0.99–1.07 | 1.03 | 0.99–1.08 |
| CD4 count > 500 | 0.76 | 0.45–1.28 | 0.69 | 0.41–1.19 | 0.70 | 0.43–1.20 | 0.64 | 0.36–1.15 |
| Undetectable viral load | 0.96 | 0.56–1.65 | 0.99 | 0.57–1.73 | 1.00 | 0.57–1.74 | 0.93 | 0.50–1.72 |

Multinomial logistic regression results (referenced to "no RLS"; n = 367).

Significance level is labeled as * $p < 0.05$, ** $p < 0.01$.

Model A: Unadjusted; Model B: demographics (age, race).

Model C: Model B + Comorbidities (anemia, depression).

Model D: Model C + medications (antidepressants).

were reported by men who had HIV < 20 years while similar prevalence of RLS was reported by men with HIV between 20–30 years and > 30 years (10.1% vs 10.2%).

## Sleep-related burden comparisons across HIV and RLS categories

On sleep questionnaires, men with HIV and RLS reported significantly worse overall sleep quality than each of the three other groups (Table 4). In addition, they reported significantly longer latency to sleep onset and time in bed than men without RLS. Finally, they were significantly sleepier than HIV negative men without RLS when comparing the ESS as a continuous variable (9.0 ± 4.6 vs 7.1 ± 4.4, p = 0.04) or at a pathologic cut off value exceeding 10 (17% vs 35%, p = 0.01). However, there were no differences in prevalence or severity of sleep disordered breathing or oxygen nadir between the groups. Actigraphic data showed that men with HIV and RLS had longer wake after sleep onset time (69 ± 30 mins vs 53 ± 26 mins, p = 0.001), lower sleep efficiency (83 ± 7% vs 87 ± 6%, p < 0.001), and greater mean nocturnal movements (22 ± 11 vs 16 ± 9, p < 0.001) than men without HIV and without RLS. The fragmentation index of men with HIV and RLS was also significantly higher than that of HIV negative men with or without RLS. Furthermore, men with HIV without RLS had significantly longer sleep onset latency, greater wake after sleep onset time, and poor sleep efficiency than men without HIV and without RLS.

## Discussion

In this cross-sectional analysis of the sleep data collected in the MACS cohort, the prevalence of RLS among men with HIV was similar to that of men without HIV. Both of these estimates fall within the RLS prevalence estimates reported for the US general population (2.5–16.0%) [1, 5, 7] and are consistent with that reported in other HIV cohorts. Employing validated questionnaire-based assessment of RLS, two prior studies have reported that the prevalence of RLS was equivalent among PLHIV and those without HIV [11, 12]. In a multisite study from the United Kingdom (N = 483; 88% White), Kunisaki et al reported the prevalence of multiple sleep disorders including RLS in older and younger individuals with HIV [12]. Older people with a median age of 60 years and HIV had a RLS prevalence of 15.5%, which was similar to the prevalence reported in the older group (14.4%) without HIV. In contrast, young

**Table 4. Comparison of sleep parameters by HIV and RLS status.**

| Characteristic | No RLS and HIV- | RLS and HIV- | No RLS and HIV + | RLS and HIV + | p-value |
|---|---|---|---|---|---|
| | (N = 330) | (N = 36) | (N = 384) | (N = 48) | |
| PSQI | | | | | |
| Total score | 6.0 ± 3.4 | 7.2 ± 3.3 | 6.8 ± 3.6 [d] | 9.3 ± 4.6 [a] | <0.001 |
| SOL, mins | 23 ± 35 | 33 ± 38 | 26 ± 27 | 42 ± 48 [b,c] | 0.002 |
| Sleep duration, hrs | 6.7 ± 1.4 | 7.0 ± 1.1 | 6.7 ± 1.3 | 6.8 ± 2.4 | 0.66 |
| Time in bed, hrs | 7.7 ± 1.2 | 8.1 ± 1.3 | 7.7 ± 1.8 | 8.6 ± 2.7 [b,c] | 0.002 |
| Epworth sleepiness scale | 7.1 ± 4.4 | 7.9 ± 3.9 | 7.3 ± 4.6 | 9.0 ± 4.6 [c] | 0.04 |
| Excessive sleepiness(%) (ESS >10) | 17 | 31 | 22 | 35 | 0.01 |
| *Polysomnography* | | | | | |
| Total sleep time, mins | 384 ± 84 | 384 ± 98 | 370 ± 88 | 354 ± 91 | 0.06 |
| AHI (events/hr) | 21 ± 18 | 22 ± 16 | 19 ± 16 | 22 ± 19 | 0.28 |
| OSA diagnosis (%) | 87 | 90 | 84 | 87 | 0.68 |
| Oxygen nadir | 83 ± 7 | 82 ± 7 | 83 ± 6 | 83 ± 8 | 0.86 |
| PLMS index | 5 ± 14 | 10 ± 16 | 5 ± 12 | 8 ± 15 | 0.07 |
| *Actigraphy* | | | | | |
| Sleep duration, mins | 412 ± 70 | 432 ± 63 | 405 ± 78 | 409 ± 75 | 0.21 |
| SOL, mins | 10 ± 9 | 10 ± 8 | 13 ± 13 [d] | 12 ± 9 | 0.009 |
| WASO, mins | 53 ± 26 | 58 ± 29 | 60 ± 29 [d] | 69 ± 30 [c] | 0.001 |
| Sleep efficiency, (%) | 87 ± 6 | 87 ± 6 | 85 ± 7 [d] | 83 ± 7 [c] | <0.001 |
| Fragmentation index | 24 ± 9 | 25 ± 9 | 26 ± 10 | 30 ± 10 [b,c] | <0.001 |
| Mean activity (count/min) | 16 ± 9 | 18 ± 11 | 19 ± 11 [d] | 22 ± 11 [c] | <0.001 |

Data presented as means ± SD or frequency (%). Group comparisons per ANOVA or Chi-square tests. PSQI Pittsburgh sleep quality index; SOL, sleep onset latency;

WASO wake after sleep onset; AHI apnea-hypopnea index; OSA, obstructive sleep apnea; PLMS periodic limb movements in sleep.

Post-hoc significance levels are labeled as follows

[a] p<0.05 for comparison between HIV+/ RLS and each of the other groups

[b] p<0.05 for comparison between HIV+/RLS and HIV+/No RLS

[c] p<0.05 for comparison between HIV+/RLS and HIV-/No RLS

[d] p<0.05 for comparisons between HIV-/No RLS and HIV+/No RLS.

participants with a median age of 46 years and HIV had a RLS prevalence of only 7.8%. Similarly, a study conducted in United States with a more diverse sample (N = 316; 41% White) also found that the prevalence of RLS was similar in people with and without HIV (11% vs 8%, p = 0.80) [11]. In contrast, a single center study (N = 228) from Germany found that the prevalence of RLS was significantly higher in people without HIV (33% vs 7%, p < 0.001) [8]. The differences across previous reports may represent differences in study sample characteristics and the methodology used for assessing RLS. The data provided here adds to the existing knowledge base about RLS in HIV by using a larger, multicenter, racially diverse cohort and examining the association of pertinent objectively-verified comorbidities, health behaviors and medications.

Although HIV status itself was not associated with RLS in adjusted analyses, several previously described demographic and medical comorbid associations were detected [1, 5, 9]. White men were twice as likely to fulfill RLS diagnostic criteria as men of minority background. These data are consistent with adjusted analyses of the Multi-Ethnic Study of Atherosclerosis where Black individuals had a lower likelihood (OR 0.56; 95% CI 0.32–0.96) of RLS with periodic limb movements as Whites [24]. Prevalence of RLS is known to be higher among individuals of Northern European background associated with genetic predisposition

[7]. Anemia was also associated with RLS given the important role of iron as a cofactor for tyrosine hydroxylase, which converts tyrosine to L-DOPA, a dopamine precursor [25]. Low peripheral iron (serum ferritin) and central nervous system stores have been associated with dopamine dysfunction and RLS severity [26–28]. Depression has been recognized to have a bidirectional relationship with RLS and recent studies have suggested that sleep disruption may partially mediate this association [29]. In a cross-sectional analysis of the Osteoporotic Fractures of Older Men study, the association between RLS severity and depressive symptoms was attenuated the most by adjusting for sleep efficiency. These data suggest that sleep onset insomnia caused by the uncomfortable sensation or sleep fragmentation caused by nocturnal periodic limb movements may precipitate depressive symptoms (e.g., fatigue, cognitive deficits). Finally, several anti-depressants (e.g., mirtazapine, venlafaxine) have also been reported to worsen or precipitate RLS by enhancing serotonergic transmission [30]. Thus, health care providers should consider these RLS risk factors when assessing PLHIV with insomnia symptoms.

A unique finding of the current study is that among men with HIV an association was noted between HIV duration and RLS. This association was independent of age, a known predictor of RLS [1]. Interestingly, all RLS cases were detectable among men with HIV duration of more than 20 years. Recently, age-related changes in the periodicity of PLMS have been reported in individuals with RLS, stabilizing during middle age but increasing after the age of 60 [31]. Whether such age-related phenomena also occur in older people with HIV is currently unknown but merits future study. Also, it is unclear if HIV-associated neurodegeneration in CNS sensory-motor modulating areas linked to RLS symptomatology (e.g., basal ganglia, substantia nigra) may potentially explain this association [26–30, 32]. As HIV may remain latent in microglia and macrophages, accelerated brain atrophy and active CNS inflammatory processes have been reported even among individuals with undetectable HIV viral load [33, 34]. Thus, a longer duration of HIV may also be associated with a chronic CNS inflammation, CNS iron deficiency, and loss of myelin integrity [27]. Recently, Hennessy et al described a higher risk of RLS among people with HIV carrying alleles for proteins regulating iron transport from glial cells and pro-inflammatory cytokine IL-17 [11]. In agreement with the data presented, some studies among people with HIV have failed to find associations between CD4 count, viral load count, ART therapy and RLS. However, Happe et al did report that CD4 count was inversely associated with RLS severity [10]. The men in MACS had been living with HIV for much longer (27 ± 12 years) than individuals in other HIV-RLS studies (e.g., Hennessy et al: 12 ± 7 years) [9]. Thus, there are numerous study specific differences which may account for these disparate associations.

This study describes for the first time the objective sleep-related burden of RLS among men with and without HIV. Men with HIV and RLS had greater sleep onset insomnia complaints, showed poorer sleep quality, greater sleepiness, and fragmented sleep than men without HIV/ RLS. Unlike many other studies in RLS, our study sample included individuals with RLS and other comorbid sleep disorders. Thus, some of the differences we detected between men with HIV and men without HIV/RLS may not be due to RLS alone. For example, although men with HIV and RLS and men without HIV/RLS had equivalent severity of OSA (i.e., AHI, oxygen nadir), it is possible that this sleep disorder may account for some of the groups' differences in levels of daytime sleepiness. A diagnosis of sleep apnea was very common across all groups indicating that multiple sleep disorders may co-exist in people with HIV and that their impact may be cumulative. For example, Kunisaki et al reported that among people with HIV, as the number of sleep disorders increased, physical, mental, and sleep-related quality of life scores worsened [12]. Furthermore, behavioral factors counterproductive to sleep consolidation (i.e., spending excessive time in bed) were more common in men with HIV and RLS than

in men without RLS. This finding may seem unusual for individuals with RLS alone as laying still in bed may activate RLS symptoms [1, 2]. However, many patients with RLS may develop comorbid insomnia, dysfunctional sleep beliefs, and engage in deleterious behaviors mimicking those of chronic insomnia sufferers (e.g., getting into bed earlier). By addressing behavioral targets and sleep cognitions, cognitive behavioral therapy for insomnia has been shown to be superior to sleep hygiene in improving insomnia, sleep quality, and sleep efficiency in people with RLS and comorbid insomnia [35]. As insomnia disorder is the most common sleep problem reported, it is of clinical value to screen PLHIV for RLS [12, 36]. Although there were several statistically significant differences in some sleep metrics detected between men without RLS by HIV status, these were smaller and likely of little clinical significance. These data show that sleep disorders are a prevalent comorbidity in HIV and represent treatment opportunities to enhance quality of life in people with HIV.

## Limitations

The current study has several important limitations. The diagnosis of RLS was established by questionnaire and did not include a clinical assessment. Although the CH-RLSq instrument we used to assess RLS is based on the diagnostic criteria, it does not provide accepted RLS diagnostic categories (i.e., indeterminate) [16]. However, these RLS categories were utilized for comparative future studies utilizing this instrument. Another major limitation of the instrument is that it does not provide an RLS severity measure akin to the international RLS study group severity rating scale [37]. Thus, it is unclear from these data if the RLS severity of men with HIV differs from that of men without HIV. Furthermore, our RLS prevalence may be underestimated by the absence of women, who have a greater prevalence of the disorder [8]. Additionally, adjustment for others lifestyle factors (e.g., stress, physical activity), nutritional deficiencies (e.g., vitamin B12), and medications (e.g., thyroxine, dopamine-blocking medications) which may impact RLS presentation was not possible [1, 8, 9]. Some association reported with RLS, although not reaching statistical significance, did have wide confidence intervals. Therefore, it is possible that other clinically important factors may be associated with RLS in HIV, which this study was under powered to detect. Finally, the method by which participants were included into the sleep ancillary protocol of the MACS may have introduced some bias.

## Conclusions

In conclusion, although the prevalence of RLS was not higher among men with HIV, it is an important diagnosis to consider among people presenting with insomnia complaints and among people with long-standing HIV. As sleep disorders are treatable and may often coexist, familiarity with the cardinal RLS symptoms and its risk factors may enhance comprehensive HIV care.

## Acknowledgments

Data in this manuscript were collected by the Multicenter AIDS Cohort Study (MACS), now the MACS/WIHS Combined Cohort Study (MWCCS). The authors gratefully acknowledge the contributions of the study participants and dedication of the staff at the MWCCS sites.

## Author Contributions

**Conceptualization:** Douglas M. Wallace, Maria L. Alcaide, Naresh M. Punjabi.

**Formal analysis:** Douglas M. Wallace, Maria L. Alcaide, Deborah L. Jones Weiss, Naresh M. Punjabi.

**Funding acquisition:** Maria L. Alcaide, Deborah L. Jones Weiss.

**Writing – original draft:** Douglas M. Wallace, Maria L. Alcaide, William K. Wohlgemuth, Deborah L. Jones Weiss, Naresh M. Punjabi.

**Writing – review & editing:** Douglas M. Wallace, Maria L. Alcaide, Deborah L. Jones Weiss, Claudia Uribe Starita, Sanjay R. Patel, Valentina Stosor, Andrew Levine, Carling Skvarca, Dustin M. Long, Anna Rubtsova, Adaora A. Adimora, Stephen J. Gange, Amanda B. Spence, Kathryn Anastos, Bradley E. Aouizerat, Yaacov Anziska.

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
