## [Decision Letter · Decision Letter 0]

19 Aug 2021

PONE-D-21-23631

Prevalence and Correlates of Restless Legs Syndrome in Men Living with HIV

PLOS ONE

Dear Dr. Wallace,

Thank you for submitting your manuscript to PLOS ONE. After careful consideration, we feel that it has merit but does not fully meet PLOS ONE’s publication criteria as it currently stands. Therefore, we invite you to submit a revised version of the manuscript that addresses the points raised during the review process.

Please note that one of the most important point, indicated by both reviewers, is that of the definition of RLS "severity"; in particular, explain if the validated IRLS scale was used or not. In the first case, it would be a strenght of the study, in the second, a very important limitations to be declared and discussed.

We look forward to receiving your revised manuscript.

Kind regards,

Raffaele Ferri

Academic Editor

PLOS ONE

Journal Requirements:

"The contents of this publication are solely the responsibility of the authors and do not represent the official views of the National Institutes of Health (NIH). MWCCS (Principal Investigators): Atlanta CRS (Ighovwerha Ofotokun, Anandi Sheth, and Gina Wingood), U01-HL146241; Baltimore CRS (Todd Brown and Joseph Margolick), U01-HL146201; Bronx CRS (Kathryn Anastos and Anjali Sharma), U01-HL146204; Brooklyn CRS (Deborah Gustafson and Tracey Wilson), U01-HL146202; Data Analysis and Coordination Center (Gypsyamber D’Souza, Stephen Gange and Elizabeth Golub), U01-HL146193; Chicago-Cook County CRS (Mardge Cohen and Audrey French), U01-HL146245; Chicago-Northwestern CRS (Steven Wolinsky), U01-HL146240; Northern California CRS (Bradley Aouizerat, Jennifer Price, and Phyllis Tien), U01-HL146242; Los Angeles CRS (Roger Detels and Matthew Mimiaga), U01-HL146333; Metropolitan Washington CRS (Seble Kassaye and Daniel Merenstein), U01-HL146205; Miami CRS (Maria Alcaide, Margaret Fischl, and Deborah Jones), U01-HL146203; Pittsburgh CRS (Jeremy Martinson and Charles Rinaldo), U01-HL146208; UAB-MS CRS (Mirjam-Colette Kempf, Jodie Dionne-Odom, and Deborah Konkle-Parker), U01-HL146192; UNC CRS (Adaora Adimora), U01-HL146194. The MWCCS is funded primarily by the National Heart, Lung, and Blood Institute (NHLBI), with additional co-funding from the Eunice Kennedy Shriver National Institute Of Child Health & Human Development (NICHD), National Institute On Aging (NIA), National Institute Of Dental & Craniofacial Research (NIDCR), National Institute Of Allergy And Infectious Diseases (NIAID), National Institute Of Neurological Disorders And Stroke (NINDS), National Institute Of Mental Health (NIMH), National Institute On Drug Abuse (NIDA), National Institute Of Nursing Research (NINR), National Cancer Institute (NCI), National Institute on Alcohol Abuse and Alcoholism (NIAAA), National Institute on Deafness and Other Communication Disorders (NIDCD), National Institute of Diabetes and Digestive and Kidney Diseases (NIDDK), National Institute on Minority Health and Health Disparities (NIMHD), and in coordination and alignment with the research priorities of the National Institutes of Health, Office of AIDS Research (OAR). MWCCS data collection is also supported by UL1-TR000004 (UCSF CTSA), UL1-TR003098 (JHU ICTR), UL1-TR001881 (UCLA CTSI), P30-AI-050409 (Atlanta CFAR), P30-AI-073961 (Miami CFAR), P30-AI-050410 (UNC CFAR), P30-AI-027767 (UAB CFAR), and P30-MH-116867 (Miami CHARM)."

"No"

 This information should be included in your cover letter; we will change the online submission form on your behalf

Reviewers' comments:

Reviewer's Responses to Questions

**Comments to the Author**

1. Is the manuscript technically sound, and do the data support the conclusions?

Reviewer #1: Yes

Reviewer #2: Partly

2. Has the statistical analysis been performed appropriately and rigorously? 

Reviewer #1: Yes

Reviewer #2: Yes

3. Have the authors made all data underlying the findings in their manuscript fully available?

Reviewer #1: Yes

Reviewer #2: Yes

4. Is the manuscript presented in an intelligible fashion and written in standard English?

Reviewer #1: Yes

Reviewer #2: Yes

5. Review Comments to the Author

Reviewer #1: Very interesting manuscript, for the first time the prevalence of RLS in patients with HIV is evaluated, on a very large sample, although (as pointed out by the authors) conducting the survey only on men represents a major limitation of the study, being RLS, as well as several immunological pathologies, more frequent in women. I believe that the authors used an excellent methodology, however, several clarifications are necessary, especially concerning the international diagnostic criteria of RLS and the disorders often related to it (PLMS), so I believe the following changes should be made:

1. On line 82 page 4 authors should refer to the international diagnostic criteria for RLS (Allen RP, Picchietti DL, Garcia-Borreguero D, Ondo WG, Walters AS, Winkelman JW, Zucconi M, Ferri R, Trenkwalder C, Lee HB; International Restless Legs Syndrome Study Group. Restless legs syndrome / Willis-Ekbom disease diagnostic criteria: updated International Restless Legs Syndrome Study Group (IRLSSG) consensus criteria - history, rationale, description, and significance. Sleep Med. 2014 Aug; 15 (8): 860-73. Doi: 10.1016 / j.sleep.2014.03.025. Epub 2014 May 17. PMID: 25023924)

2. On line 85 page 4 authors should refer to the international diagnostic criteria for periodic movements of the lower limbs in sleep (Ferri R, Fulda S, Allen RP, Zucconi M, Bruni O, Chokroverty S, Ferini-Strambi L, Frauscher B, Garcia-Borreguero D, Hirshkowitz M, Högl B, Inoue Y, Jahangir A, Manconi M, Marcus CL, Picchietti DL, Plazzi G, Winkelman JW, Zak RS; International and European Restless Legs Syndrome Study Groups (IRLSSG and EURLSSG) . World Association of Sleep Medicine (WASM) 2016 standards for recording and scoring leg movements in polysomnograms developed by a joint task force from the International and the European Restless Legs Syndrome Study Groups (IRLSSG and EURLSSG). Sleep Med. 2016 Oct; 26: 86-95. Doi: 10.1016 / j.sleep.2016.10.010. Epub 2016 Nov 7. PMID: 27890390).

3. Line 146 page 6: it is advisable to specify the cutoff values of the ESS scale to be considered pathological.

4. Line 155 page 7: the international diagnostic criteria for PLMS must be indicated in the bibliographic notes (already indicated in point 2), then replace in the bibliographic note 17.

5. Line 255 page 12: the authors speak of "definite or indeterminate RLS", a denomination that however does not exist in the diagnostic criteria and in the definition of the pathology.

6. Line 358 page 17: Add as another limitation of the study the fact that the authors did not also consider the severity of RLS, for example by means of the international Restless Legs Syndrome study group severity rating scale:

a. Walters AS, LeBrocq C, Dhar A, Hening W, Rosen R, Allen RP, Trenkwalder C. Validation of the International Restless Legs Syndrome Study Group rating scale for restless legs syndrome. The International Restless Legs Syndrome Study Group. Sleep Med. 2003 Mar;4(2):121-32

b. Sharon D, Allen RP, Martinez-Martin P, Walters AS, Ferini Strambi L, Högl B, Trotti LM, Buchfuhrer M, Swieca J, Bogan RK, Zak R, Hensley JG, Schaefer LA, Marelli S, Zucconi M, Stefani A, Holzknecht E, Olvera V, Meaklim H , Laska I, Becker PM; International RLS Study Group. Validation of the self-administered version of the international Restless Legs Syndrome study group severity rating scale - The sIRLS. Sleep Med. 2019 Feb; 54: 94-100. Doi: 10.1016 / j.sleep.2018.10.014. Epub 2018 Oct 29. PMID: 30529783.), another important feature to evaluate, besides the duration of illness.

7. Line 360 page 17: add that also the distribution of PLMS varies with age in patients with RLS (Ferri R, DelRosso LM, Silvani A, Cosentino FII, Picchietti DL, Mogavero P, Manconi M, Bruni O. Peculiar lifespan changes of periodic leg movements during sleep in restless legs syndrome. J Sleep Res. 2020 Jun; 29 (3): e12896. doi: 10.1111 / jsr.12896. Epub 2019 Jul 16. PMID: 31313413) and therefore it would be interesting to evaluate this aspect in patients with HIV, also in the light of what is described in the discussion also on PLMS.

Reviewer #2: In the paper “Prevalence and correlates of Restless Legs Syndrome in men living with HIV” the authors aim to determine the prevalence of RLS, associated clinical correlates, and characterize sleep-related differences in men with and without HIV. They included 942 men from the Multicenter AIDS Cohort 107 Study (MACS). The prevalence of definite RLS was comparable in men with and without HIV. White race, anemia, depression, and antidepressant use were each independently associated with RLS. HIV disease duration was also associated with RLS. Men with HIV and RLS reported poorer sleep quality, greater sleepiness, and had worse objective sleep efficiency/fragmentation than men without HIV/RLS. The study describes RLS features among men with and without HIV, although some limitations described by the authors themselves are present. Indeed, to be included in this analysis, participants had to complete the sleep questionnaires, overnight polysomnography, and actigraphy.

It could be interesting to know if RLS symptoms were chronic or intermittent adding informations on the frequency of RLS symptoms in patients with and without HIV.

The authors found an interesting correlation between HIV disease duration and RLS. In these cases, has the patients been diagnosed before the study and asking for treatment?

In the discussion session the authors reported increased RLS severity in patients with HIV in comparison with patients without HIV. What does it mean “severity” if any RLS severity scale has not been included in the study?

Men with HIV and RLS had greater sleepiness, but once again the lack of excessive daytime time sleepiness is a supportive criteria for RLS diagnosis. The authors reported that sleep apnea was very common across all groups indicating that multiple sleep disorders may co-exist in people with HIV and that their impact may be cumulative. This aspect is interesting but it should be distinguished the impact of other sleep disorders from the impact of RLS itself.

Finally, I suggest to discuss some unexpected findings. For example, the authors reported a longer time in bed among patients with RLS. That is really strange because bed is usually the worst place for the onset of RLS symptoms.

6. PLOS authors have the option to publish the peer review history of their article (what does this mean?). If published, this will include your full peer review and any attached files.

Reviewer #1: No

Reviewer #2: No

---

## [Author Response · Author response to Decision Letter 0]

6 Sep 2021

09-04-2021

Dear Dr Ferri: 

 We thank you and the reviewers of our manuscript for the thoughtful feedback. Please find enclosed a point-by-point rebuttal to the reviewers’ comments. 

Reviewer 1

1. On line 82 page 4 authors should refer to the international diagnostic criteria for RLS (Allen RP, Picchietti DL, Garcia-Borreguero D, Ondo WG, Walters AS, Winkelman JW, Zucconi M, Ferri R, Trenkwalder C, Lee HB; International Restless Legs Syndrome Study Group. Restless legs syndrome / Willis-Ekbom disease diagnostic criteria: updated International Restless Legs Syndrome Study Group (IRLSSG) consensus criteria - history, rationale, description, and significance. Sleep Med. 2014 Aug; 15 (8): 860-73. Doi: 10.1016 / j.sleep.2014.03.025. Epub 2014 May 17. PMID: 25023924)

This reference has been added as suggested.

2. On line 85 page 4 authors should refer to the international diagnostic criteria for periodic movements of the lower limbs in sleep (Ferri R, Fulda S, Allen RP, Zucconi M, Bruni O, Chokroverty S, Ferini-Strambi L, Frauscher B, Garcia-Borreguero D, Hirshkowitz M, Högl B, Inoue Y, Jahangir A, Manconi M, Marcus CL, Picchietti DL, Plazzi G, Winkelman JW, Zak RS; International and European Restless Legs Syndrome Study Groups (IRLSSG and EURLSSG) . World Association of Sleep Medicine (WASM) 2016 standards for recording and scoring leg movements in polysomnograms developed by a joint task force from the International and the European Restless Legs Syndrome Study Groups (IRLSSG and EURLSSG). Sleep Med. 2016 Oct; 26: 86-95. Doi: 10.1016 / j.sleep.2016.10.010. Epub 2016 Nov 7. PMID: 27890390).

This reference has been added as suggested.

3. Line 146 page 6: it is advisable to specify the cutoff values of the ESS scale to be considered pathological.

This detail (ESS > 10) has been added in the Methods section as suggested. This metric has also been added to Table 4.

4. Line 155 page 7: the international diagnostic criteria for PLMS must be indicated in the bibliographic notes (already indicated in point 2), then replace in the bibliographic note 17.

This has been changed as suggested.

5. Line 255 page 12: the authors speak of "definite or indeterminate RLS", a denomination that however does not exist in the diagnostic criteria and in the definition of the pathology.

We have clarified that this RLS terminology is per the developer of the questionnaire and does not exist in the RLS diagnostic criteria. This has been explicated fully in the Methods and Limitations sections. However, we specifically retained this terminology because we solely utilized this questionnaire to determine the RLS categories. We have removed the language concerning RLS “diagnostic categories” to read RLS “categories”

6. Line 358 page 17: Add as another limitation of the study the fact that the authors did not also consider the severity of RLS, for example by means of the international Restless Legs Syndrome study group severity rating scale:

a. Walters AS, LeBrocq C, Dhar A, Hening W, Rosen R, Allen RP, Trenkwalder C. Validation of the International Restless Legs Syndrome Study Group rating scale for restless legs syndrome. The International Restless Legs Syndrome Study Group. Sleep Med. 2003 Mar;4(2):121-32.

We have added this to the Limitations subsection of the Discussion and added references as suggested.

7. Line 360 page 17: add that also the distribution of PLMS varies with age in patients with RLS (Ferri R, DelRosso LM, Silvani A, Cosentino FII, Picchietti DL, Mogavero P, Manconi M, Bruni O. Peculiar lifespan changes of periodic leg movements during sleep in restless legs syndrome. J Sleep Res. 2020 Jun; 29 (3): e12896. doi: 10.1111 / jsr.12896. Epub 2019 Jul 16. PMID: 31313413) and therefore it would be interesting to evaluate this aspect in patients with HIV, also in the light of what is described in the discussion also on PLMS.

We have added this interesting point to the discussion and highlighted this potential future aim.

Reviewer 2

1. It could be interesting to know if RLS symptoms were chronic or intermittent adding informations on the frequency of RLS symptoms in patients with and without HIV.

One of the items of the CH-RLSq queries RLS symptom frequency over the last 12 months. We have added details about the response pattern to this item to the results section.

2. The authors found an interesting correlation between HIV disease duration and RLS. In these cases, has the patients been diagnosed before the study and asking for treatment?

There was no information collected during the MACS sleep ancillary study on whether individuals had a pre-existing diagnosis of RLS. The medication categories collected were broad (i.e., antidepressant, tranquilizers) making the adjustment for more RLS-specific (e.g., dopamine agonists) medications was not possible. 

3. In the discussion session the authors reported increased RLS severity in patients with HIV in comparison with patients without HIV. What does it mean “severity” if any RLS severity scale has not been included in the study?

We have removed mention of “severity” of RLS throughout the manuscript. As has been correctly pointed out, the score of the CH-RLSq cannot be inferred to mean greater RLS severity. The lack of severity assessment of the CH-RLSq has been added as a major limitation of the instrument in the Limitations subsection of the Discussion.

4. Men with HIV and RLS had greater sleepiness, but once again the lack of excessive daytime time sleepiness is a supportive criteria for RLS diagnosis. The authors reported that sleep apnea was very common across all groups indicating that multiple sleep disorders may co-exist in people with HIV and that their impact may be cumulative. This aspect is interesting but it should be distinguished the impact of other sleep disorders from the impact of RLS itself.

The reviewer is correct in pointing out that most patient with RLS patients may not report excessive daytime sleepiness (reference). In fact, Table 4 now shows that 31% of men without HIV and RLS and 35% of men with HIV and RLS had pathologic levels of daytime sleepiness (ESS > 10). However, an important difference between our study and many studies exploring daytime sleepiness in RLS is that most have exclude individuals with comorbid sleep disorders (Kallweit U et al 2009; Fulda S et al 2007). Therefore, some of the sleepiness may originate from these other comorbid sleep issues or other comorbidities that may be associated with hypersomnia and RLS (i.e., depressive symptoms). We have highlighted this important difference in our sample in the Discussion. 

5. Finally, I suggest to discuss some unexpected findings. For example, the authors reported a longer time in bed among patients with RLS. That is really strange because bed is usually the worst place for the onset of RLS symptoms.

This is a good suggestion for clarification. Many patients with RLS have comorbid insomnia and engage in behaviors that further impact their sleep adversely. By targeting these behavioral targets, a recent RCT showed the superiority of cognitive behavioral therapy for insomnia over sleep hygiene in improving sleep quality, efficiency, and insomnia symptoms in patients with RLS and comorbid insomnia (Song ML et al Sleep Med 2020 Oct;74:227-234). We have added this reference and point to the discussion.

Once again, thank you for the gracious comments. We believe these changes have strengthened the manuscript considerably. 

Sincerely,

Douglas Wallace and colleagues.

---

## [Decision Letter · Decision Letter 1]

20 Sep 2021

Prevalence and correlates of restless legs syndrome in men living with HIV

PONE-D-21-23631R1

Dear Dr. Wallace,

We’re pleased to inform you that your manuscript has been judged scientifically suitable for publication and will be formally accepted for publication once it meets all outstanding technical requirements.

Kind regards,

Raffaele Ferri

Academic Editor

PLOS ONE

Additional Editor Comments (optional):

Reviewers' comments:

Reviewer's Responses to Questions

**Comments to the Author**

1. If the authors have adequately addressed your comments raised in a previous round of review and you feel that this manuscript is now acceptable for publication, you may indicate that here to bypass the “Comments to the Author” section, enter your conflict of interest statement in the “Confidential to Editor” section, and submit your "Accept" recommendation.

Reviewer #1: (No Response)

Reviewer #2: All comments have been addressed

2. Is the manuscript technically sound, and do the data support the conclusions?

Reviewer #1: (No Response)

Reviewer #2: Yes

3. Has the statistical analysis been performed appropriately and rigorously? 

Reviewer #1: (No Response)

Reviewer #2: Yes

4. Have the authors made all data underlying the findings in their manuscript fully available?

Reviewer #1: (No Response)

Reviewer #2: Yes

5. Is the manuscript presented in an intelligible fashion and written in standard English?

Reviewer #1: (No Response)

Reviewer #2: Yes

6. Review Comments to the Author

Reviewer #1: (No Response)

Reviewer #2: The authors have adequately addressed all commnents including some changes throughout the text. The data support the conclusions

7. PLOS authors have the option to publish the peer review history of their article (what does this mean?). If published, this will include your full peer review and any attached files.

Reviewer #1: No

Reviewer #2: No

---

## [Editor Report · Acceptance letter]

23 Sep 2021

PONE-D-21-23631R1 

Prevalence and correlates of restless legs syndrome in men living with HIV 

Dear Dr. Wallace:

I'm pleased to inform you that your manuscript has been deemed suitable for publication in PLOS ONE. Congratulations! Your manuscript is now with our production department. 

Kind regards, 

on behalf of

Dr. Raffaele Ferri 

Academic Editor

PLOS ONE